# Rampant tooth loss across 200 million years of frog evolution

**Daniel J Paluh[1,2]\*, Karina Riddell[1], Catherine M Early[1,3], Maggie M Hantak[1], Gregory FM Jongsma[1], Rachel M Keeffe[1,2], Fernanda Magalhães Silva[1,4], Stuart V Nielsen[1], María Camila Vallejo-Pareja[1,2], Edward L Stanley[1], David C Blackburn[1]**

[1]Department of Natural History, Florida Museum of Natural History, University of Florida, Gainesville, United States; [2]Department of Biology, University of Florida, Gainesville, United States; [3]Biology Department, Science Museum of Minnesota, Saint Paul, United States; [4]Programa de Pós Graduação em Zoologia, Universidade Federal do Pará, Museu Paraense Emilio Goeldi, Belém, Brazil

**Abstract** Teeth are present in most clades of vertebrates but have been lost completely several times in actinopterygian fishes and amniotes. Using phenotypic data collected from over 500 genera via micro-computed tomography, we provide the first rigorous assessment of the evolutionary history of dentition across all major lineages of amphibians. We demonstrate that dentition is invariably present in caecilians and salamanders, but teeth have been lost completely more than 20 times in frogs, a much higher occurrence of edentulism than in any other vertebrate group. The repeated loss of teeth in anurans is associated with a specialized diet of small invertebrate prey as well as shortening of the lower jaw, but it is not correlated with a reduction in body size. Frogs provide an unparalleled opportunity for investigating the molecular and developmental mechanisms of convergent tooth loss on a large phylogenetic scale.

\*For correspondence:
dpaluh@ufl.edu

**Competing interests:** The authors declare that no competing interests exist.

## Introduction

The evolution of teeth is considered a key innovation that promoted the radiation of jawed vertebrates, facilitating the transition from a passive to active predatory lifestyle (*Gans and Northcutt, 1983*). Teeth are complex mineralized tissues that originated in stem gnathostomes more than 400 million years ago (*Rücklin et al., 2012*) and have been broadly maintained across living chondrichthyans, actinopterygians, and sarcopterygians due to the critical role these structures play in the acquisition and processing of food. The shape, size, location, and number of teeth differ widely across vertebrates, especially in response to broad variation in food type. Although dentition is generally conserved across vertebrates, teeth have been lost completely several times, resulting in toothlessness or edentulism, including in three extant clades of mammals (baleen whales, anteaters, and pangolins), turtles, and birds (*Davit-Béal et al., 2009*). Teeth are likely lost following the evolution of a secondary feeding tool that improves the efficiency of food intake (e.g., beak, baleen, specialized tongue), leading to relaxed functional constraints on dentition (*Davit-Béal et al., 2009*). In contrast to other tetrapods, the evolution and diversity of teeth in amphibians has been poorly studied, despite long recognition that frogs—one of the most diverse vertebrate orders with more than 7000 species—possess variation in the presence or absence of teeth.

All living salamanders and caecilians are assumed to have teeth on the upper jaw, lower jaw, and palate (*Duellman and Trueb, 1986*), but nearly all frogs lack dentition on the lower jaw and variably possess teeth on the upper jaw and palate. When present, amphibian teeth are typically pedicellate (each tooth consists of a crown and pedicel separated by a noncalcified dividing zone), bicuspid, homodont, and continuously replaced (*Davit-Béal et al., 2007*). Recent work suggests that dentition

on the lower jaw was lost in the ancestor of frogs more than 200 million years ago and was subsequently regained in a single species (*Gastrotheca guentheri*; *Boulenger, 1882*) during the Miocene (*Wiens, 2011*). The presence or absence of dentition has also been considered an important taxonomic character in frogs; for example, a subclass was once proposed that included all toothless species (Bufoniformia; *Cope, 1867*). Our understanding of the anuran tree of life has fundamentally changed with the development of molecular phylogenetics (*Duellman and Trueb, 1986*; *Feng et al., 2017*; *Hime et al., 2021*), but there has been no attempt to estimate the frequency of tooth loss across frog diversity or evaluate the factors that may be correlated with edentulism. Most frogs are generalist, gape-limited predators that capture prey using tongue propulsion (*Regal and Gans, 1976*), reducing the importance of teeth in prey capture. Tooth loss is hypothesized to occur in frogs that specialize on eating small prey (microphagy), such as ants and termites (*Das and Coe, 1994*; *Parmelee, 1999*; *Narváez and Ron, 2013*). This may lead to relaxed functional constraints on energetically expensive teeth. Microphagous frogs are known to have shortened jaws and altered feeding cycles (*Emerson, 1985*), modified tongues (*Trueb and Gans, 1983*), and some have the ability to sequester dietary alkaloids from their prey, rendering them toxic (*Caldwell, 1996*; *Vences et al., 1998*). Alternately, teeth may be reduced or lost as a byproduct of miniaturization or truncated development (paedomorphosis; *Davies, 1989*; *Hanken and Wake, 1993*; *Smirnov and Vasil'eva, 1995*) because the initiation of odontogenesis occurs ontogenetically late in frogs (during or after metamorphosis) compared to other vertebrates.

Using the most recent species-rich phylogeny of extant amphibian species (*Jetz and Pyron, 2018*) and our extensive taxonomic sampling via high-resolution X-ray micro-computed tomography (microCT) of over 500 of the 562 currently recognized amphibian genera (*Amphibia-Web, 2021*), we (1) evaluated the phylogenetic distribution of teeth and reconstructed the evolutionary history of dentition across all major lineages of amphibians and (2) tested whether dietary specialization, relative jaw length, and body size are correlated with the loss of teeth in frogs. Our results demonstrate that the presence and location of teeth are highly conserved in salamanders and caecilians, but labile in frogs. We found that teeth have been repeatedly lost in frogs and at a much higher frequency than in any other vertebrate group. The evolution of edentulism in anurans is correlated with a microphagous diet and shortening of the lower jaw but not with a reduction in body size over evolutionary time. Six reversals, from edentulous to toothed jaws, were inferred in frogs.

## Results

### Distribution of teeth in amphibians

We recorded the presence or absence of teeth on each dentigerous bone of the lower jaw, upper jaw, and palate for 524 amphibian species (*Figure 1*; Dataset S1). Taxa were coded as 'toothed' if teeth were observed on any cranial element and 'edentulous' if teeth were entirely absent. Our survey of amphibian dentition across the majority of extant genera confirmed that all salamanders and caecilians retain teeth, while 134 of the 429 frog species examined are entirely edentulous (Dataset S1). All anuran species lack dentary teeth with the exception of *G. guentheri*. The evolution of maxillary and premaxillary teeth of the upper jaw is synchronized in all frog species (*Figure 1A and B*), being present in 292 taxa and coordinately absent in 136 species. The vomerine teeth on the palate are the most variable in frogs, being present in 202 species and absent in 226. Many anurans have maxillary and premaxillary teeth in the absence of vomerine teeth (92 species), but only two species examined have vomerine teeth while lacking upper jaw teeth (*Rhombophryne testudo*, *Uperodon systoma*). No other anuran cranial elements possess teeth.

All 65 salamander species examined have teeth on the lower jaw and palate, but three species lack upper jaw teeth on the maxilla and premaxilla (the sirenids *Siren intermedia* and *Pseudobranchus striatus* and the salamandrid *Salamandrina terdigitata*). *Thorius pennatulus* (Plethodontidae) and two proteids (*Necturus lewisi* and *Proteus anguinus*) lack maxillary teeth but retain premaxillary teeth. All salamanders have vomerine teeth on the palate (including the paravomerine tooth patches that underlie the parasphenoid in plethodontids; *Figure 1C*; *Lawson et al., 1971*). Palatal teeth were additionally observed on the palatopterygoid (*N. lewisi* and *P. anguinus*) and palatine (*S. intermedia* and *P. striatus*). The lower jaw teeth are present on the dentary in all species, except *S.*

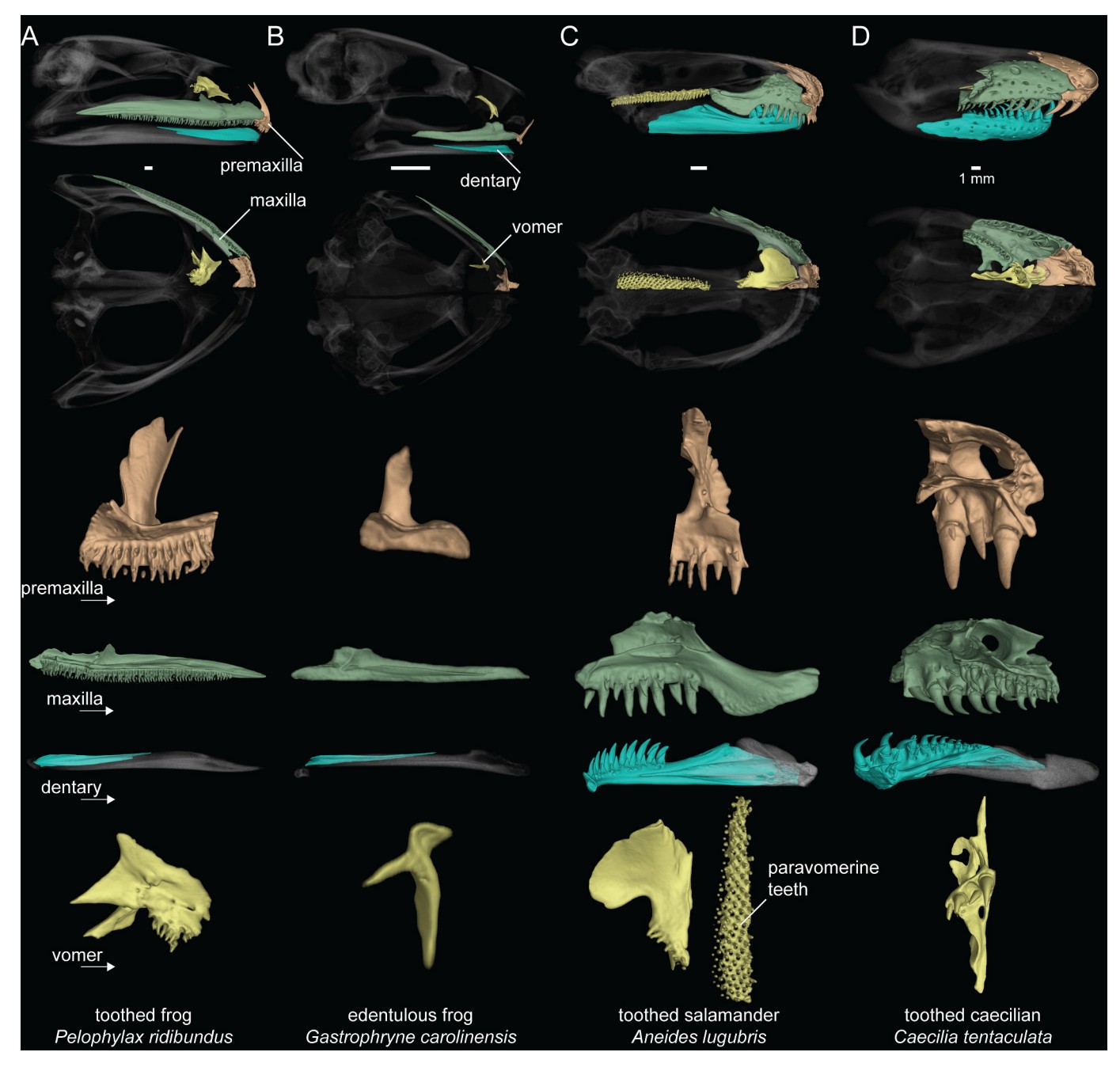

**Figure 1.** Dental diversity of amphibians. (A) Toothed frog, *Pelophylax ridibundus* (CAS:Herp:217695), (B) edentulous frog, *Gastrophryne carolinensis* (UF:Herp:110645), (C) toothed salamander, *Aneides lugubris* (MVZ:Herp:249828), (D) toothed caecilian, *Caecilia tentaculata* (KU:Kuh:175441). Skulls in lateral and ventral views: dentigerous cranial elements are colored and the remainder of the skull is semi-transparent. Isolated premaxilla (orange), maxilla (green), dentary (blue), and vomer (yellow) in lingual views. Teeth are present on all colored elements except the dentary in *P. ridibundus* and those of *G. carolinensis*. Scale bars = 1 mm.

*intermedia* and *P. striatus*, which have mandibular teeth on the splenial. *Necturus lewisi* and *P. anguinus* are the only two species that have lower jaw teeth on both the dentary and splenial.

All 30 caecilian species examined possess marginal teeth on the lower and upper jaw and palatal teeth on multiple elements (*Figure 1D*). The individual elements of the lower jaw in caecilians fuse to form the pseudodentary, and this composite element varies in having either one or two rows of

teeth. Upper jaw teeth are present on the nasopremaxilla (fused nasal and premaxilla) and maxillo-palatine (fused maxilla and palatine; outer row). Palatal teeth are always present on the vomer and maxillopalatine (inner row) and occur on the ectopterygoid in one species (*Geotrypetes seraphini*).

## Repeated tooth loss in frogs

Teeth are absent in 134 anuran genera belonging to 19 families. We used reversible-jump Markov chain Monte Carlo (MCMC) in RevBayes (*Höhna et al., 2016*) to conduct ancestral state reconstructions and sample all five Markov models of phenotypic character evolution in proportion to their posterior probability for toothed and edentulous states in 524 amphibian species. The maximum a posteriori model of dentition evolution was the one-rate model with a posterior probability of 0.91. The model-averaged maximum a posteriori ancestral state of Lissamphibia and Anura is toothed with a posterior probability of 0.99. Teeth have been completely lost at least 22 times in frogs (*Figure 2*), and six reversals from edentulous to toothed upper jaws were inferred. Edentulism has evolved three times in Archaeobatrachia (in Pipoidea, *Feng et al., 2017*) and 19 times in Neobatrachia (10 times in Hyloidea, six times in Ranoidea, twice in Myobatrachidae, and once in Nasikabatrachidae). One reversal was estimated in Myobatrachidae (in *Uperoleia mahonyi*; *Clulow et al., 2016*) and five reversals were inferred in Microhylidae (in *Dyscophus, Uperodon, Anodonthyla, Cophyla*, and *Rhombophryne + Plethodontohyla*).

We compared six discrete character evolution models using fitMk in phytools (*Revell, 2012*) for three dental states (fully toothed [on premaxilla, maxilla, vomer]; toothed upper jaw [premaxilla, maxilla] with vomerine tooth loss; edentulous) in 425 anuran species to test if vomerine tooth loss precedes complete edentulism in frogs. The all-rates-different model was the best fit (all-rates-different Akaike information criterion (AIC) weight (AICw) = 0.91, equal-rates AICw = 0.00, single-rate ordered AICw = 0.00, symmetric ordered AICw = 0.01, unsymmetric ordered AICw = 0.02, symmetric unordered AICw = 0.06), indicating that vomerine tooth loss does not always precede complete edentulism (*Figure 3*). Stochastic character mapping suggests that the number of transitions to edentulism is similar between toothed upper jaws with toothless vomers (13.8) and fully toothed states (10.3). Toothed frogs have lost vomerine teeth an estimated 59 times (*Figure 3*). One gain of vomerine teeth subsequent to a loss was inferred in *Phlyctimantis leonardi*.

## Relationships among tooth loss, diet, and body size

We compiled published diet records for 268 frog lineages and classified 69 taxa from 20 families as microphagous (defined here as >50% of diet by number or volume consisting of ants, termites, or mites) and 199 taxa from 47 families as generalist feeders (*Figure 4*; Dataset S2). Of the 69 microphagy specialists, 54 are edentulous and 15 are toothed. Of the 199 generalists, 25 are edentulous and 174 are toothed. A BayesTrait discrete analysis indicated correlated evolution between edentulism and microphagy: the dependent model of trait evolution is strongly supported over the independent model (Bayes factor = 54.32; a Bayes factor >2 implies the evolution of two traits is linked). Similar results were found using a 155-taxon dataset excluding genus-level diet data (Bayes factor = 26.28) and a 134-taxon dataset excluding genus-level diet data and species-level diet data based on a small sample size (less than five individuals; Bayes factor = 16.0). Of the 22 independent losses of teeth across frogs, at least 16 of these lineages contain microphagous species (*Figure 4*). The majority of the 25 taxa classified as both edentulous and generalist feeders are a subset of the bufonids (N = 14) and microhylids (N = 5) sampled, but also includes the fully aquatic pipids *Pipa* and *Hymenochirus*, two brevicipitids (*Probreviceps* and *Callulina*), and the Darwin's frog, *Rhinoderma*.

Head and body measurements were recorded for 423 anuran species. The relative jaw length in frogs ranges from 62% of head length in *Synapturanus mirandaribeiroi*, an edentulous microhylid, to 140% of head length in *Lepidobatrachus asper*, a toothed ceratophryid. A phylogenetic logistic regression showed a significant relationship between edentulism and shortened jaws (alpha = 0.0011, standard error = 0.8920, p<0.001; *Figure 5A*). Edentulous species have an average relative jaw length of 83% of head length, while toothed species have an average relative jaw length of 99% of head length. Nearly all edentulous species examined have an anteriorly shifted jaw joint (narrow-mouthed due to a lower jaw length that is shorter than respective head length; the two largest bufonids in our dataset, *Bufo gargarizans* and *Rhaebo blombergi*, are exceptions), but over 100 toothed

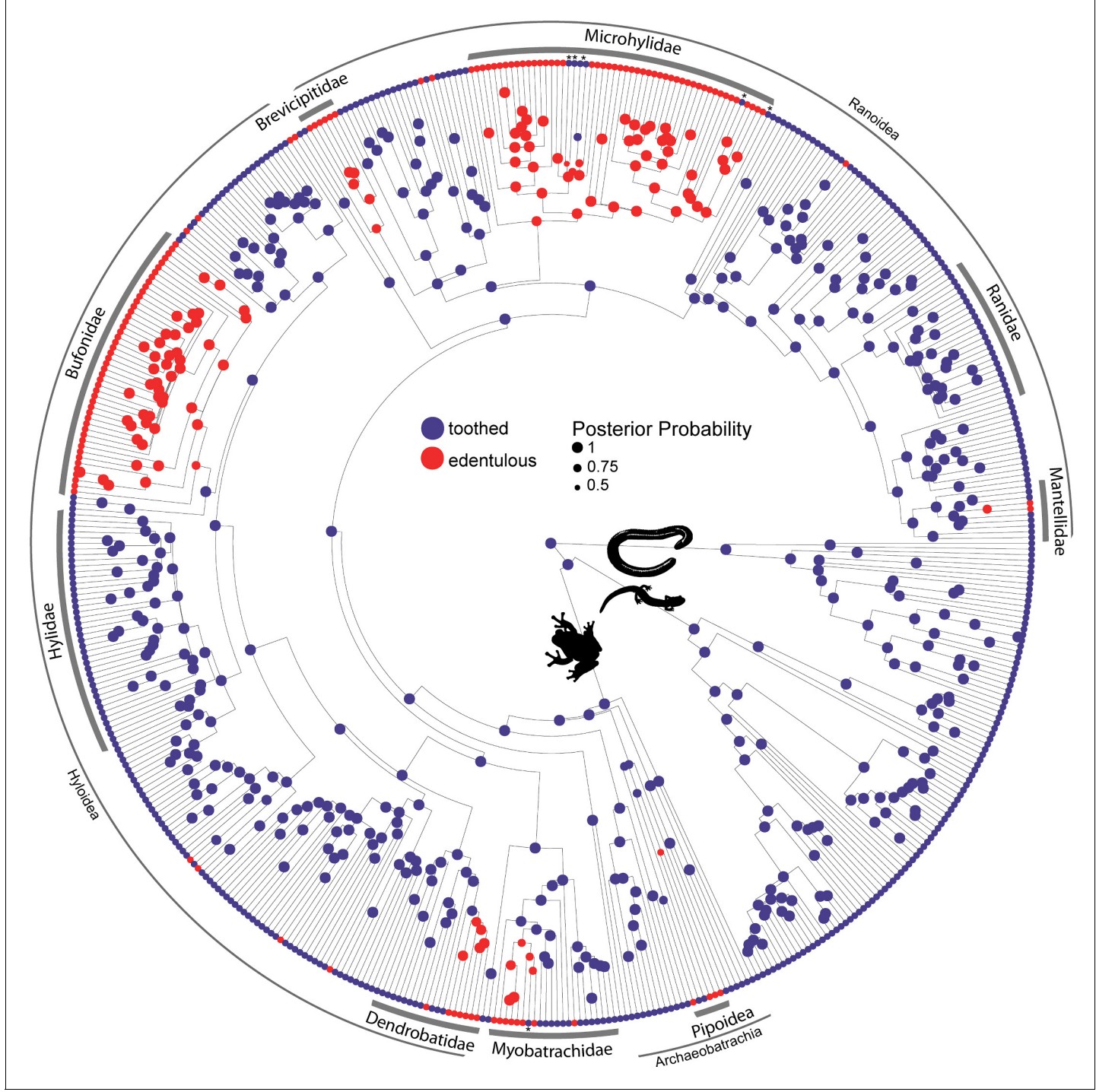

**Figure 2.** Phylogeny of 524 amphibians depicting the evolution of dentition. Node point color corresponds to Bayesian model-averaged ancestral states of dentition: blue = toothed; red = edentulous. The size of each node point represents the posterior probability of the most probable ancestral state. Tip point colors correspond to dentition states for all species. Asterisks indicate inferred reversals. For species tip labels display *Figure 2—figure supplement 2*. Corresponding data are provided in Dataset S1.

The online version of this article includes the following figure supplement(s) for figure 2:

**Figure supplement 1.** Comparison of true teeth, odontoids, and dental anatomy of the microhylid taxa with inferred evolutionary reversals.
**Figure supplement 2.** Phylogeny of 524 amphibians depicting the evolution of dentition with species tip labels.
**Figure supplement 3.** Evolution of dentary teeth.
**Figure supplement 4.** Evolution of premaxillary teeth.

*Figure 2 continued on next page*

**Figure supplement 5.** Evolution of maxillary teeth.
**Figure supplement 6.** Evolution of vomerine teeth.

taxa have posteriorly shifted jaws (lower jaws that are longer than their heads; *Figure 5A*). The snout–vent length (SVL) of specimens measured ranges from 7.8 mm in the edentulous *Paedophryne amauensis*, the smallest known frog, to 263.9 mm in the toothed *Conraua goliath*, the largest known extant species. A phylogenetic logistic regression indicated that there is no relationship between edentulism and body size (alpha = 0.0016, standard error = 0.1162, p=0.09; *Figure 5B*). Edentulous species have an average SVL of 36.2 mm (range 7.8–152.4 mm) and toothed species have an average SVL of 43.7 (range 11.2–263.9 mm). However, a relationship was found between edentulism and body size when excluding bufonids (alpha = 0.0020, standard error = 0.1539, p=0.015), a clade representing a single loss of dentition but that varies widely in size (11.7–152.4 mm SVL in our dataset).

## Discussion

### Evolution of edentulism in jawed vertebrates

With at least 22 independent origins of edentulism, frogs have completely lost teeth more times than any other vertebrate clade. Based on our review of the literature, only eight other extant vertebrate lineages are entirely edentulous. There are no described edentulous chondrichthyan species (but see *Mulas et al., 2020* for the first described aberrant case in a catshark). To our knowledge, teeth have been entirely lost only three times in living actinopterygian fishes in the (1) Gonorynchiformes excluding Gonorynchidae (*Kohno et al., 1996*; *Britz and Moritz, 2007*), (2) Gyrinocheilidae

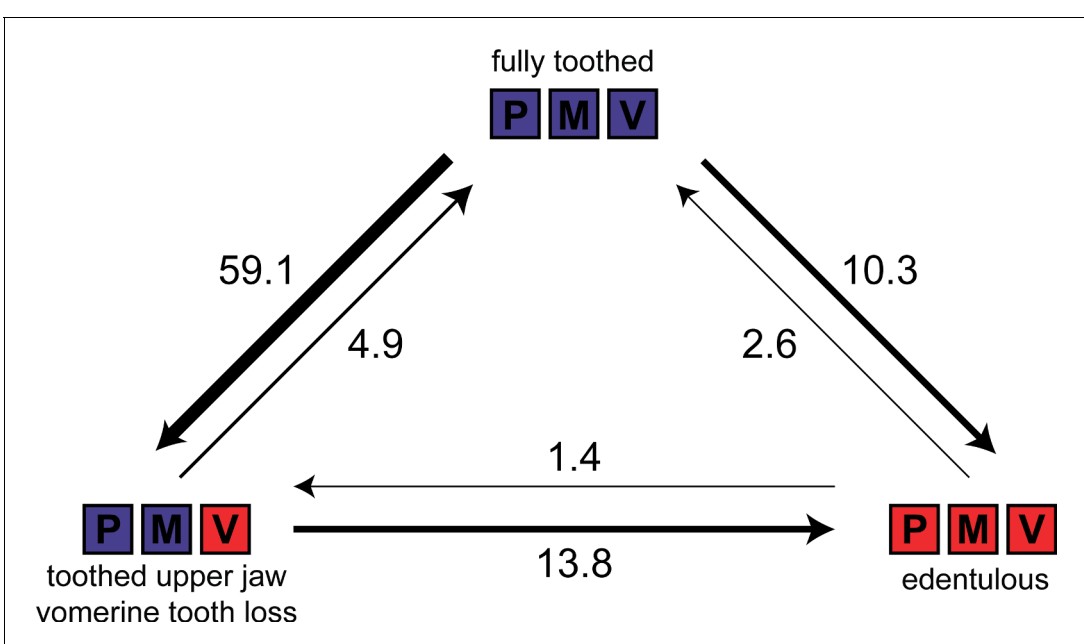

**Figure 3.** Estimated number of evolutionary transitions among three dental states (fully toothed, toothed upper jaw with vomerine tooth loss, edentulous) inferred from stochastic character mapping using 1000 replicates. P = premaxilla, M = maxilla, V = vomer. Width of arrows corresponds to estimated number of changes.

The online version of this article includes the following figure supplement(s) for figure 3:

**Figure supplement 1.** Discrete character evolution model comparisons for three dental states (fully toothed, toothed upper jaw with vomerine tooth loss, edentulous) in 425 species of frogs.
**Figure supplement 2.** Ancestral reconstruction of three dental character states (fully toothed, toothed upper jaw with vomerine tooth loss, edentulous) using stochastic character mapping in 425 species of frogs.

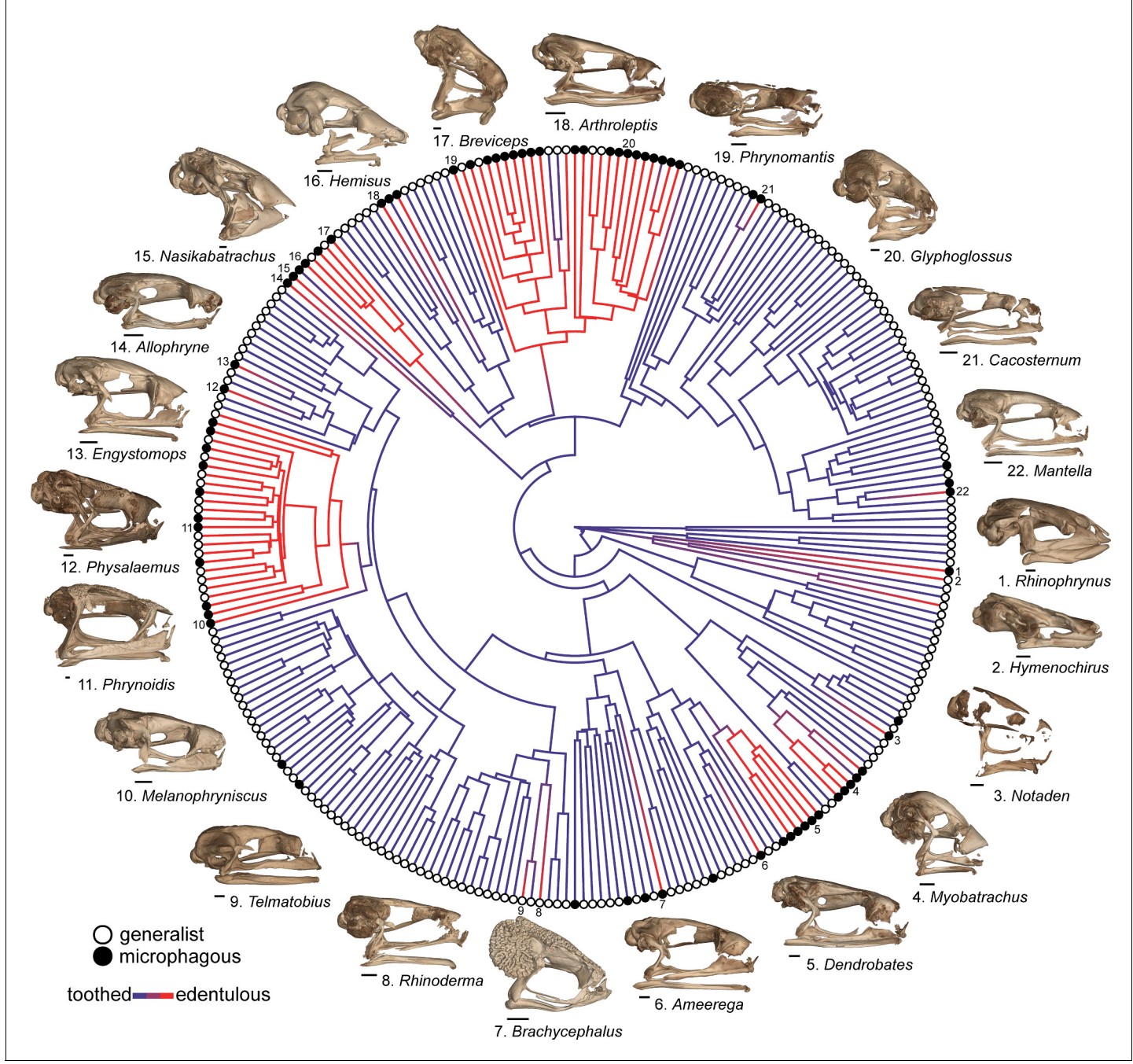

**Figure 4.** Phylogeny of 268 frog species with a stochastic character map of dentition states and distribution of generalist and microphagous diet states (tip point colors) illustrating the correlated evolution of edentulism and microphagy. Diversity of edentulous frog skulls: 1. *Rhinophrynus dorsalis* (CAS: Herp:71766), 2. *Hymenochirus boettgeri* (CAS:Herp:253587), 3. *Notaden bennetti* (CAS:Herp:78115), 4. *Myobatrachus gouldii* (MCZ:Herp:A-139543), 5. *Dendrobates tinctorius* (YPM:VZ:HERA 016010), 6. *Ameerega trivittata* (UF:Herp:107200), 7. *Brachycephalus ephippium* (UF:Herp:72725), 8. *Rhinoderma darwinii* (UF:Herp:62022), 9. *Telmatobius carrillae* (UF:Herp:39717), 10. *Melanophryniscus stelzneri* (UF:Herp:63183), 11. *Phrynoidis asper* (USNM: Amphibians and Reptiles:586870), 12. *Physalaemus nattereri* (MCZ:Herp:A30113), 13. *Engystomops pustulosus* (CAS:SUA:21892), 14. *Allophryne ruthveni* (KU:Kuh:166716), 15. *Nasikabatrachus sahadryensis* (CES:F:203), 16. *Hemisus guineensis* (CAS:Herp:258533), 17. *Breviceps gibbosus* (AMNH: Herpetology:3053), 18. *Arthroleptis schubotzi* (CAS:Herp:201762), 19. *Phrynomantis annectens* (UF:Herp:187273), 20. *Glyphoglossus molossus* (CAS: Herp:243121), 21. *Cacosternum namaquense* (CAS:Herp:156975), 22. *Mantella baroni* (CAS:Herp:250387). Scale bars = 1 mm. For species tip labels display *Figure 4—figure supplement 1*. Corresponding data are provided in Dataset S2.

The online version of this article includes the following figure supplement(s) for figure 4:

**Figure supplement 1.** Phylogeny of 268 frog species with a stochastic character map of dentition and the distribution of diet states with species tip labels.

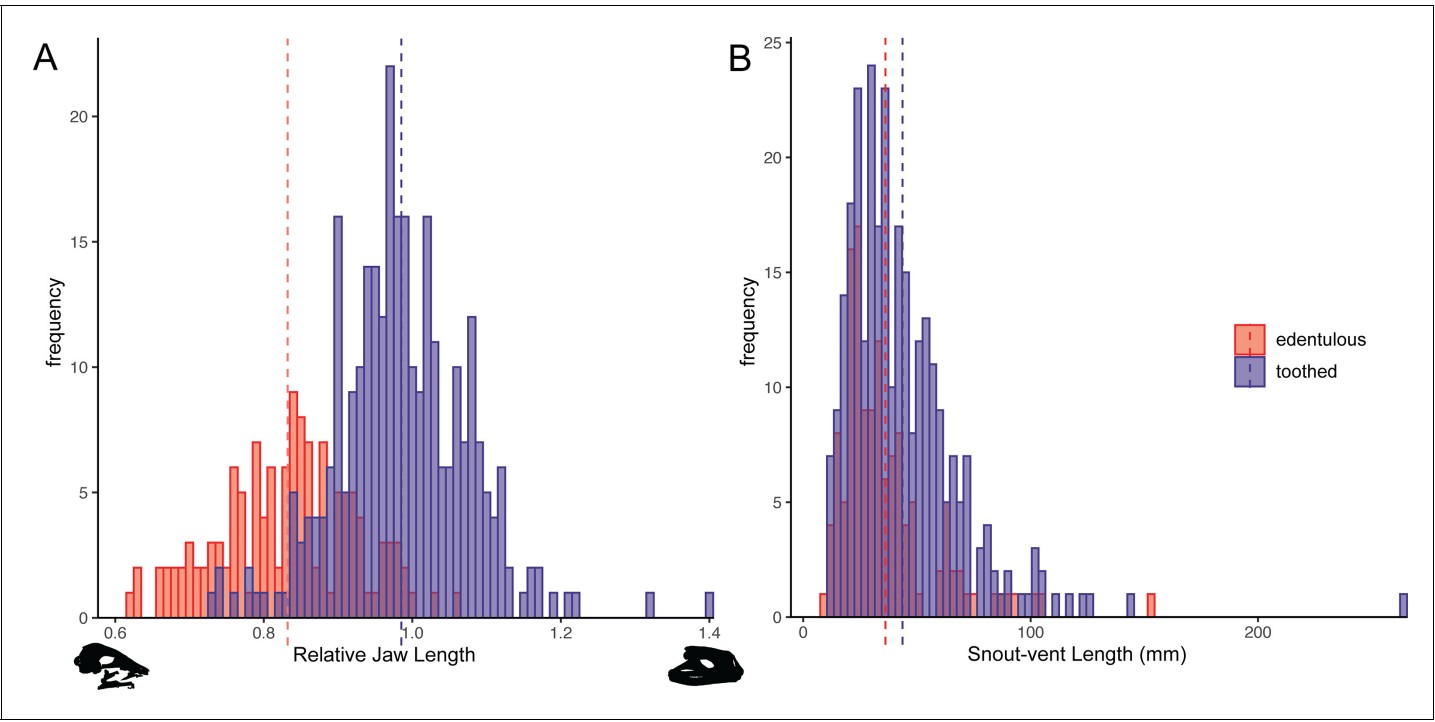

**Figure 5.** Histograms of relative jaw length (mandible length divided by skull length; **A**) and body size (snout–vent length [SVL]; **B**) in 423 frog species plotted by dentition states (blue = toothed; red = edentulous). A phylogenetic correlation was identified between tooth loss and shortened lower jaws. There is no association between edentulism and body size. Left skull silhouette is *Hemisus guineensis* (CAS:Herp:258533) and right skull is *Lepidobatrachus asper* (UF:Herp:12347). Corresponding trait data are provided in Dataset S3.

(*Conway, 2011*), and (3) Syngnathidae (seahorses and pipefish; *Lin et al., 2016*). Other fish lineages, such as the cyprinids, have toothless oral jaws but retain true pharyngeal teeth (*Aigler et al., 2014*). Five extant amniote clades are edentulous, including three lineages of mammals (baleen whales, pangolins, anteaters), all living birds, and all living turtles (Davit-Béal et al., 2009). There are several mammal clades that have lost enamel but retain reduced teeth (armadillos, sloths, aardvarks, pygmy, and dwarf sperm whales; *Meredith et al., 2009*). Molecular evidence suggests a single loss of teeth in the common ancestor of extant birds (*Meredith et al., 2014*), but complete edentulism also evolved independently in at least two extinct lineages of Mesozoic birds (*Confuciusornis* and *Gobipteryx*; *Yang and Sander, 2018*; *Brocklehurst and Field, 2021*). Teeth were completely lost in at least two lineages of non-avian dinosaurs (ornithomimosaurs and caenagnathoids; *Wang et al., 2017*; *Hendrickx et al., 2019*) and in some pterosaurs, such as members of Azhdarchidae (*Yang and Sander, 2018*). All living crocodilians retain teeth, but at least two fossil suchian archosaurs were edentulous (*Shuvosaurus* and *Effigia*; *Nesbitt and Norell, 2006*). There are no known edentulous squamate species, although African egg-eating snakes in the genus *Dasypeltis* may have a dental polymorphism, as they typically have small, short teeth but some individuals are reported to be edentulous (*Visser, 1981*). *Emydocephalus*, a genus of sea snakes that specializes on eating fish eggs, has maxillary fangs and pterygoid teeth but has lost teeth on the palatine, premaxilla, and lower jaw (*Voris, 1966*). Lastly, at least one extinct rhynchocephalian has been suggested to be edentulous (*Sapheosaurus*, *Rauhut et al., 2012*).

The loss of teeth may be associated with the evolution of a secondary feeding apparatus (*Davit-Béal et al., 2007*; *Wang et al., 2017*), such as the keratinized beak in birds and turtles, baleen in mysticete whales, and specialized tongues in pangolins and anteaters. Nearly all frogs have a specialized tongue that is used in feeding (*Regal and Gans, 1976*), and this adaptation might have facilitated the repeated loss of teeth across anurans. Surprisingly, three anuran genera are both tongueless and edentulous (*Hymenochirus*, *Pseudhymenochirus*, and *Pipa* in Pipidae), but these species are highly aquatic and have a derived mechanism of catching prey under water through suction feeding (*Dean, 2003*; *Cundall et al., 2017*). The edentulous syngnathids (seahorses and relatives)

and actinopterygians with toothless oral jaws also catch prey through suction feeding (*Roos et al., 2009*; *Mihalitsis and Bellwood, 2019*). There appears to be no size-related constraints promoting complete tooth loss across all vertebrates. Edentulous species span the entire spectrum of vertebrate body sizes: the smallest known vertebrate species (the microhylid frog *P. amauensis*, *Rittmeyer et al., 2012*) and the largest (the blue whale, *Balaenoptera musculus*) are both edentulous. The second-smallest known vertebrate, the cyprinid fish in the genus *Paedocypris*, retains true pharyngeal teeth (*Kottelat et al., 2006*). Several edentulous vertebrate clades are thought to have paedomorphic skulls, including toothless frogs (*Smirnov and Vasil'eva, 1995*), birds (*Bhullar et al., 2012*), and baleen whales (*Fordyce and Barnes, 1994*), suggesting that truncated ontogenetic trajectories may constrain the formation of teeth, but this hypothesis requires further investigation.

Tooth formation occurs ontogenetically late in frogs, during or after metamorphosis, in contrast to during early larval or embryonic development in other vertebrates (*Davit-Béal et al., 2007*, *Lainoff et al., 2015*). This delayed shift in odontogenesis may be linked to the evolutionary lability of teeth in anurans. There may also be a relationship between the loss of teeth and delayed ossification of dentigerous elements. For example, the dentary bone ossifies relatively late in frogs, and nearly always lacks teeth, compared to being one of the first cranial elements to ossify in salamanders and caecilians (*Harrington et al., 2013*), and these amphibians always retain mandibular dentition. Truncated development is hypothesized to be associated with the repeated loss of other mineralized structures in frogs, such as the stapes of the middle ear (*Pereyra et al., 2016*; *Womack et al., 2018*; *Womack et al., 2019*). The anuran mouth undergoes dramatic restructuring during metamorphosis while transitioning from an herbivorous tadpole with a keratinized beak and short, cartilaginous lower jaw to a carnivorous frog with an elongated, bony lower jaw. This rapid morphological transformation requires further study in edentulous and toothed species. Several anuran lineages have evolved direct development (undergoing the larval stage within the egg; *Gomez-Mestre et al., 2012*), and this life history transition may provide an opportunity to repattern the jaw and alter dental development, such as the timing of tooth germ initiation. Many of the edentulous frogs identified in this study are biphasic (possessing a free-swimming tadpole stage), but edentulism is also present in some direct-developing lineages (e.g., *Arthroleptis*, *Brachycephalus*, Brevicipitidae, *Myobatrachus*, asterophryine microhylids) and viviparous species (*Nimbaphrynoides*, *Nectophrynoides*), suggesting that life history mode does not constrain patterns of tooth loss.

## Amphibian dentition and tooth loss in frogs

Dentition is highly conserved in salamanders and caecilians with no identified cases of edentulism. Teeth are present on the jaws and palate of all caecilians (*Wake and Wurst, 1979*), and this is also the typical dental condition in salamanders (*Gregory et al., 2016*). The aquatic sirenid salamanders (*Siren*, *Pseudobranchus*) lack maxillary and premaxillary teeth (*Clemen and Greven, 1988*), while the miniaturized species of *Thorius* sampled here (*T. pennatulus*) lacks maxillary teeth but retains teeth on the premaxilla, palate, and lower jaw. At least one species of *Thorius* possesses a novel dental polymorphism in which males lack maxillary teeth but females maintain several teeth on the maxilla (*Hanken et al., 1999*). To our knowledge, this is the only known case of a sexually dimorphic presence or absence dental polymorphism in an amphibian. Larval salamanders and caecilians were excluded in our dentition survey but differ in patterns of dentition from adults (*Wake, 1976*; *Clemen and Greven, 2018*), such as the transient presence of teeth on the splenials, palatines, and pterygoids that are lost during development (*Schoch et al., 2019*). Maxillary and premaxillary teeth are synchronized in all anuran taxa that we sampled, but two species in the genus *Telmatobius* have maxillary teeth in the absence of premaxillary teeth (*Barrionuevo, 2017*). Members of this genus can be toothed or edentulous, and two species are reported to have intraspecific variation in the presence or absence of dentition (*Barrionuevo, 2017*). The loss of vomerine teeth is not a prerequisite for complete tooth loss (*Figure 3*), and the presence or absence of teeth on the vomer is not coordinated with dentition on the maxilla and premaxilla in frogs. These results suggest that dentition on the upper jaws and palate are independent modules, and the higher lability of vomerine teeth requires further study. Previous work has suggested that the size and number of vomerine teeth may be correlated with diet and body size (*Hedges, 1989*; *Estrada and Hedges, 1996*). Teeth are entirely absent in 134 anuran genera distributed across 19 families, and our ancestral state reconstruction suggests that teeth have been lost more than 20 times during the evolution of frogs.

We identified a phylogenetic correlation between the evolution of edentulism and a microphagous diet, and these two traits co-occur in more than 50 genera belonging to 14 families (Dataset S2; *Figure 4*). The majority of these species specialize on eating ants and termites, despite that these insects have many defense behaviors (biting, stinging, chemical weapons) and low nutritional value compared to other invertebrates (*Redford and Dorea, 1984*; *McNab, 1984*). Edentulous, microphagous frogs inhabit biomes ranging from tropical forests (e.g., *Dendrobates*, *Mantella*, *Cardioglossa*) to arid deserts (e.g., *Breviceps*, *Notaden*) and are found on all continents, excluding Antarctica. Frogs, ants, and termites evolved at roughly the same time—with important diversification events occurring in all three groups during the Cretaceous and Cretaceous–Paleogene boundary (*Moreau and Bell, 2013*; *Bourguignon et al., 2015*; *Feng et al., 2017*)—suggesting the repeated evolution of complete edentulism in frogs may be linked to the spatiotemporal diversification of ants and termites. Teeth have been repeatedly reduced in other tetrapods that specialize on eating ants and termites, including multiple lineages of mammals (echidnas, numbats, aardvarks, aardwolves, anteaters, armadillos, pangolins; *Reiss, 2001*) and squamates (scolecophidian blind snakes, *Aprasia* worm lizards; *Daza and Bauer, 2015*).

The complete loss of teeth in frogs is associated with the shortening of the lower jaw (*Figure 4*), a skeletal trait that is known to occur in species that eat smaller prey (*Emerson, 1985*; *Vidal-García and Scott Keogh, 2017*; *Paluh et al., 2020*). The shortening of the mandible reduces maximum gape and alters jaw biomechanics to improve the efficiency of catching many small prey items. Frogs with a jaw length equal to or longer than the skull have an asymmetrical feeding cycle where the time spent catching prey is short but the time spent bringing prey into the mouth is long (*Gans and Gorniak, 1982*); shortened jaws result in a faster, symmetric feeding cycle where equal amounts of time are spent catching and bringing prey into the mouth (*Emerson, 1985*). At least four lineages of edentulous anurans that specialize on ants and termites have additionally evolved muscular hydrostatic tongues that can be aimed in all three dimensions and with great precision without moving the head to improve the efficiency of small prey capture (*Rhinophrynus*, *Trueb and Gans, 1983*; *Hemisus*, *Nishikawa et al., 1999*; microhylids and brevicipitids, *Meyers et al., 2004*). Recently, these same species have additionally been shown to have relatively small eyes among frogs (*Thomas et al., 2020*), suggesting reduced visual sensitivity and supporting the hypothesis that microphagous frogs may possess improved abilities to process olfactory and tactical cues in order to detect and localize prey (*Deban et al., 2001*). Therefore, the evolution of microphagy may have influenced a suite of traits in the head of frogs, including skull shape, jaw position, dentition, tongue morphology, and the functionality of sensory organs.

The majority of the 134 edentulous frogs in our dataset are restricted to the families Bufonidae and Microhylidae. All 48 genera of bufonids examined—the only anuran clade widely recognized as being edentulous (*Davit-Béal et al., 2007*)—and 48 of 54 of microhylid genera examined lack teeth. All remaining families have less than 10 edentulous genera. The Bufonidae and Microhylidae are two of the most diverse frog families, comprising 637 and 697 species, respectively (18% of all frogs; *AmphibiaWeb, 2021*). The evolution of edentulism in frogs may exert an influence on diversification rates, but we refrain from testing this hypothesis using trait-dependent diversification models due to our sparse, genus-level taxonomic sampling (429 tips representing 7299 lineages). The results of our ancestral state reconstruction analyses indicate that teeth were independently lost in the most recent common ancestors of both bufonids and microhylids. Once lost, teeth have not been regained in the Bufonidae but may have re-evolved several times in microhylids. Although both clades have many taxa that specialize on small prey, there are bufonids and microhylids with expanded, generalized diets, and a few species that will even consume vertebrate prey (e.g., *Rhinella marina*, *Asterophrys turpicola*). The variation in diet within bufonids and microhylids corresponds with variation in the relative length of the lower jaw (Dataset S3) and overall skull morphology (*Paluh et al., 2020*).

The inferred reversals in Microhylidae occur in *Dyscophus*, *Uperodon*, and four cophyline genera (*Anodonthyla*, *Cophyla*, *Plethodontohyla*, *Rhombophryne*). Recent work has also suggested that additional microhylid genera contain toothed and edentulous species (*Mini*, *Scherz et al., 2019*; *Glyphoglossus*, *Gorin et al., 2021*). If teeth were entirely lost in the common ancestor of microhylids, the repeated re-evolution of true teeth (with enamel, dentin, and pulp cavity) in this clade is unlikely and requires histological investigation. We hypothesize the tooth-like structures in some of these taxa, such as *Rhombophryne* and *Uperodon* (*Figure 2—figure supplement 1*), may be small odontoids (bony serrations that lack the tissue composition of true teeth; *Fabrezi and Emerson, 2003*),

similar to what has been described in the bufonid *Incilius alvarius* (*Mendelson and Pramuk, 1998*), some *Brachycephalus* (*Ribeiro et al., 2017*), and some New Guinea asterophrynine microhylids (*Zweifel, 1971*). Alternatively, the most recent common ancestor of the Microhylidae possessed teeth and several lineages then subsequently lost dentition independently, with no reversals. The dental anatomy of *Dyscophus* has been examined (*LaDouceur et al., 2020*), and this genus does possess true teeth. The morphology of *Dyscophus*, *Anodonthyla*, *Cophyla*, and *Plethodontohyla* indicates that these taxa have true dentition based on the presence of replacement teeth and/or floating teeth that are undergoing resorption (*Figure 2—figure supplement 1*). The phylogenetic relationships and divergence time estimates among microhylid taxa remain controversial (*Peloso et al., 2016*; *Feng et al., 2017*; *Streicher et al., 2020*), which further impedes the interpretation of dental evolution in this group.

Of the ten anuran genera known to possess variation in the presence or absence of teeth, diet data are only available for *Physalaemus* and *Engystomops*. In both genera, edentulous species have specialized microphagous diets in comparison to toothed congeners that consume a broader array of invertebrates (*Narváez and Ron, 2013*). Dietary alkaloid sequestration has evolved as a predator defense mechanism in at least five clades of frogs that specialize on eating ants and mites, and teeth have been lost in several of these lineages (Dendrobatinae and *Ameerega* [Dendrobatidae, *Saporito et al., 2004*]; *Pseudophryne* [Myobatrachidae, *Smith et al., 2002*]; *Mantella* [Mantellidae, *Daly et al., 1997*]; *Melanophryniscus* [Bufonidae, *Hantak et al., 2013*]). Teeth are retained in *Phyllobates*, which is sister to all other edentulous genera of the Dendrobatinae, and in the Cuban *Eleutherodactylus* group known to sequester alkaloids (*Rodríguez et al., 2011*). There are other microphagous frogs that retain teeth, such as the ant specialist *Sphaenorhynchus* (*Parmelee, 1999*). Further work is needed to investigate the number, size, and histological anatomy of teeth across toothed frogs that vary in diet. It remains unknown whether any anurans have lost enamel but retain teeth, which has occurred several times in mammals (*Meredith et al., 2009*).

No relationship was identified between complete edentulism and body size in the 423 frog species sampled. The smallest known species of frog, *P. amauensis*, lacks teeth, but some miniaturized anurans are toothed. We examined 25 taxa with an SVL of 15 mm or less: 13 were toothed and 12 were edentulous. Several of the smallest edentulous species in our dataset are microhylids, bufonids, and dendrobatids, and these clades have widespread tooth loss across a range of body sizes. We identified only one case of edentulism in Brachycephaloidea, in the genus *Brachycephalus*, despite this neotropical radiation of over 1000 species containing many miniaturized lineages (e.g., smallest members of *Pristimantis*, *Eleutherodactylus*, *Noblella*). Within the genus *Arthroleptis*, several miniature species lack teeth (*Laurent, 1954*; *Blackburn, 2008*), suggesting that, in some cases, a reduction in body size and tooth loss may be linked. There are several large or gigantic species within the Bufonidae (*Womack and Bell, 2020*), but all true toads lack teeth regardless of size. When excluding bufonids, reduction in body size was phylogenetically correlated with edentulism, indicating that further work is needed to investigate the interplay between edentulism, body size evolution, and paedomorphism across anurans.

## Tooth loss in fossil amphibians

To our knowledge, no stem tetrapods have been described as edentulous (*Ruta et al., 2003*; *Anderson et al., 2008*; *Matsumoto and Evans, 2017*). Albanerpetontids, an extinct lineage of lissamphibians, had teeth on the premaxilla, maxilla, and dentary (*Daza et al., 2020*), and there is no evidence of edentulism in fossil salamanders or caecilians. Of the stem salientians with cranial material, teeth are present on the upper jaw in *Prosalirus* (*Shubin and Jenkins, 1995*), *Vieraella* (*Báez and Basso, 1996*), and *Liaobatrachus* (*Gao and Wang, 2001*). No teeth are visible in *Triadobatrachus* (*Ascarrunz et al., 2016*), the oldest known stem frog, but the maxilla and premaxilla are poorly preserved in this impression fossil. The dentary of *Triadobatrachus* lacks teeth, and the absence of dentition on the lower jaw is considered a synapomorphy of Salientia (*Milner, 1988*).

Several extinct crown-group anuran genera are edentulous, and the majority of these are hypothesized to be members of the Pipidae (*Báez and Harrison, 2005*), including *Saltenia* from the late Cretaceous of Argentina (*Báez, 1981*), *Vulcanobatrachus* from the late Cretaceous of South Africa (*Trueb et al., 2005*), *Shelania* from the Paleocene-Eocene of Argentina (*Báez and Trueb, 1997*, *Báez and Pugener, 1998*), and *Singidella* from the Eocene of Tanzania (*Báez and Harrison, 2005*). Based on the phylogenetic analyses of *Báez and Harrison, 2005* and *Trueb et al., 2005*, extinct

edentulous pipids likely represent two or more instances of tooth loss: at least once in the South American fossils (*Saltenia*, *Shelania*; sister to African *Xenopus*; clade Xenopodinomorpha) and once in the African fossils (*Vulcanobatrachus*, *Singidella*; sister to South American *Pipa* and African *Hymenochirus* and *Pseudhymenochirus*; clade Pipinomorpha). However, the clades Xenopodinomorpha and Pipinomorpha are not supported by molecular phylogenetic analyses of living pipids (*Irisarri et al., 2011*; *Feng et al., 2017*), and therefore the placement of these fossils requires re-evaluation. Tooth loss likely occurred in pipids at earliest during the mid-Cretaceous because all described pipimorph frogs from the early Cretaceous are toothed (*Báez et al., 2021*): *Thoraciliacus* (*Trueb, 1999*), *Cordicephalus* (*Trueb and Báez, 2006*), *Neusibatrachus*, and *Gracilibatrachus* (*Báez, 2013*). Our ancestral state reconstructions suggest that living pipid frogs have lost teeth twice independently in the ancestor of *Hymenochirus* + *Pseudhymenochirus* and in *Pipa* (*Figure 2*). All species of *Xenopus* and three of seven species of *Pipa* have teeth (*P. carvalhoi*, *P. aspera*, *P. arrabali*; *Trueb and Cannatella, 1986*). Teeth may have been lost several times in *Pipa*, but the relationships among species are poorly understood.

The sister clade to Pipidae is Rhinophrynidae (collectively forming the Pipoidea, *Feng et al., 2017*), which contains a single living species, the edentulous *Rhinophrynus dorsalis*. The fossil record of Rhinophrynidae is sparse (*Blackburn et al., 2019*) but includes the toothed *Rhadinosteus* from the late Jurassic of Utah (*Henrici, 1998*) and the edentulous *Chelomophrynus* from the Eocene of Wyoming (*Henrici, 1991*). The presence of teeth in *Rhadinosteus* supports the hypothesis that teeth were lost independently in pipids and rhinophrynids (*Figure 2*) and indicates that edentulism may have evolved later in rhinophrynids.

Two edentulous, non-pipoid fossil frogs have been described from the Mesozoic of the Northern Hemisphere: *Theatonius* from the late Cretaceous of Wyoming (*Fox, 1976*) and *Tyrrellbatrachus* from the late Cretaceous of Alberta (*Gardner, 2015*). These two crown-group anurans have an uncertain placement in the frog tree of life but are likely distantly related to one another. *Gardner, 2015* hypothesized that these two taxa likely represent two independent cases of tooth loss and possibly the oldest record of edentulism in nonpipid frogs. The majority of living, edentulous frogs are neobatrachians (*Figure 2*; in Hyloidea, Ranoidea, Myobatrachidae, and Nasikabatrachidae), but few fossil neobatrachians have been described as edentulous (*Gardner, 2015*).

## Molecular and developmental mechanisms of tooth loss

Recent work has documented that several lineages of edentulous vertebrates have various states of molecular tooth decay in the genes that are critical for the formation of dentin and enamel with frameshift mutations and stop codons that result in nonfunctionalization (mammals: *Meredith et al., 2009*; turtles: *Meredith et al., 2013*; birds: *Meredith et al., 2014*; syngnathids: *Lin et al., 2016*). The frameshift mutation rate of these loci can be used to estimate the timing of tooth loss in the fossil record (*Meredith et al., 2009*; *Meredith et al., 2014*), and the ratio of synonymous and nonsynonymous substitutions can be calculated to measure selection pressure on enamel matrix proteins (*Alazem and Abramyan, 2019*). Whether edentulous frogs possess similar rates of molecular tooth decay in these loci, as demonstrated in amniotes, has yet to be tested. Recently, *Lu et al., 2021* failed to identify any dentin or enamel genes in the genomes of two bufonids (*R. marina* and *B. gargarizans*) but *Shaheen et al., 2021* demonstrated pseudogenization of the major tooth enamel gene amelogenin in *R. marina*, *B. gargarizans*, and *B. bufo*. Based on selection intensity estimates and substitution rates, this gene likely became inactivated in the Bufonidae 40–60 million years ago (*Shaheen et al., 2021*). We hypothesize that these tooth-specific genes have degenerated repeatedly across edentulous anurans by novel inactivating mutations, and the frameshift mutation rate will indicate that teeth were lost at several different geologic times during the evolution of frogs. Anuran enamel matrix proteins may be operating under relaxed selection, compared to purifying selection in most mammals and reptiles (*Alazem and Abramyan, 2019*), due to the evolution of projectile tongue feeding, enabling the evolutionary lability of frog teeth.

The developmental genetics of tooth formation in amphibians is almost entirely unexplored, especially when compared to our understanding of chondrichthyan, teleost, and amniote odontogenesis (*Fraser et al., 2004*; *Tucker and Sharpe, 2004*; *Thiery et al., 2017*). It is unknown if the genes critical for tooth formation in fishes and amniotes are also expressed during morphogenesis of teeth in amphibians, if all frog species retain a suppressed ancestral developmental pathway of tooth development on the lower jaw, or if the odontogenetic pathway has been disrupted via one or

many mechanisms on the jaws of edentulous anurans. The loss of teeth on the lower jaw of frogs could be due to the loss of a single major signal that can orchestrate odontogenesis, comparable to the sole loss of odontogenic *Bmp4* expression in living birds (*Chen et al., 2000*) or termination of *Msx2* expression in living turtles (*Tokita et al., 2013*), which arrests tooth formation early in development. If true, potential rudimentary structures, such as tooth buds or the early thickening of the odontogenic band, might be seen before the abortion of tooth development in the lower jaw of anurans. Investigation of the developmental genetics of tooth formation in the upper and lower jaws of frogs will fill a large gap in our understanding of vertebrate evolution and development and may elucidate the mechanisms of repeated tooth loss and putative cases of the re-evolution of lost teeth in one of the most diverse vertebrate orders.

## Materials and methods

### Species sampling and scanning

We collected data from high-resolution microCT scans of 523 amphibian species, representing 420 frog genera (of 460 total; *AmphibiaWeb, 2021*), 65 salamander genera (of 68 total), and 30 caecilian genera (of 34 total). One recently described frog species was not microCT scanned but included in the dataset because it is the only member of its genus with teeth (*U. mahonyi*; *Clulow et al., 2016*). All genera are represented by one species except for nine anuran genera (*Arthroleptis*, *Cacosternum*, *Engystomops*, *Gastrotheca*, *Physalaemus*, *Pipa*, *Telmatobius*, *Uperodon*, *Uperoleia*) with two sampled lineages that represent known dental variation within these genera (Dataset S1). All scans were run using a 240 kv X-ray tube containing a diamond-tungsten target, with the voltage, current, and detector capture time adjusted for each scan to maximize absorption range for each specimen. Raw X-ray data were processed using GE's proprietary datos|x software version 2.3 to produce a series of tomogram images and volumes, with final voxel resolutions ranging from 1 to 147 μm. The resulting microCT volume files were imported into VG StudioMax version 3.2.4 (Volume Graphics, Heidelberg, Germany), the skull and skeleton were isolated using VG StudioMax's suite of segmentation tools, and then exported as high-fidelity mesh files. We deposited image stacks (TIFF) and 3D mesh files of the skull and skeleton for each specimen in MorphoSource (see Dataset S1 for DOIs).

### Survey of amphibian dentition variation and ancestral state reconstructions

We recorded the presence or absence of teeth on each dentigerous bone of the lower jaw, upper jaw, and palate for 524 amphibian species (*Figure 1*; Dataset S1). Teeth were identified by a combination of the following characteristics: conical shape, presence of distinct pedicel and crown, presence of replacement teeth, and/or presence of floating teeth that are undergoing resorption (*Figure 1*, *Figure 2—figure supplement 1*). We conducted ancestral state reconstructions of dentition (two states: toothed, edentulous) in extant amphibians using the data collected from 524 species representing 515 genera and all 77 families using the phylogeny of *Jetz and Pyron, 2018*. Bayesian ancestral state reconstructions were calculated using reversible-jump MCMC in RevBayes (*Höhna et al., 2016*) to sample all five Markov models of phenotypic character evolution (one-rate, two-rate, zero-to-one irreversible, one-to-zero irreversible, no change) in proportion to their posterior probability. We accounted for model uncertainty by making model-averaged ancestral state estimates (*Freyman and Höhna, 2018*; *Freund et al., 2018*). The models were assigned an equal prior probability using a uniform set-partitioning prior, and the root state frequencies were estimated using a flat Dirichlet prior. The rates of gain and loss of dentition were drawn from an exponential distribution with a mean of 10 expected character state transitions over the tree. The MCMC was run for 22,000 iterations, the first 2000 iterations were discarded as burn-in, and samples were logged every 10 iterations. Convergence of the MCMC was confirmed using Tracer version 1.6 to ensure that analyses had reached stationarity. We conducted additional Bayesian ancestral state reconstructions using RevBayes to model the evolutionary history of dentition presence or absence on individual dentigerous elements (*Figure 2—figure supplements 3–6*).

To test if vomerine tooth loss always precedes complete edentulism in frogs, we compared discrete character evolution models using fitMk in phytools for three dental states (fully toothed, toothed upper jaw with vomerine tooth loss, edentulous) in 425 species of frogs. The only two

species that putatively possess vomerine teeth while lacking upper jaw teeth (*R. testudo* and *U. systoma*) were excluded from this analysis. Six models were compared (equal-rates, single-rate ordered, symmetric ordered, unsymmetric ordered, symmetric unordered, all-rates-different) using the AIC and AICw (*Figure 3—figure supplement 1*). The ordered models enforced vomerine tooth loss as an intermediate state between fully toothed and edentulous states. To estimate the posterior probabilities for all nodes and the number of transitions between these three character states, we used stochastic character mapping (MCMC sampling of character histories from their posterior probability distribution; *Huelsenbeck et al., 2003*) in phytools (make.simmap function) using the best fit model of character evolution (all-rates-different), a flat root prior, and 1000 replicates (*Figure 3—figure supplement 2*).

## Testing relationships among edentulism, diet, and body size

We compiled dietary data for all sampled anuran species from the literature (see Dataset S2 for references). Species were classified as microphagous specialists if the majority (>50%) of their diet by number or volume consists of ants, termites, or mites. Species were classified as generalists if the majority of their diet by number or volume consists of other invertebrate groups or vertebrates. For species with no published diet records, we searched for any existing diet records at the genus level because dentition state (toothed/edentulous) is generally consistent within a genus. Only 10 of 461 anuran genera are known to contain both edentulous and toothed species (*Arthroleptis*, *Cacosternum*, *Engystomops*, *Glyphoglossus*, *Mini*, *Physalaemus*, *Pipa*, *Telmatobius*, *Uperodon*, *Uperoleia*). Due to the disparity in existing ecological data available across all anurans, the dietary records ranged from singular reports (one prey item in one individual) to detailed studies investigating the stomach contents of hundreds of individuals.

We measured SVL (tip of the snout to the rear of the ischium), skull length (occiput to tip of the snout), and mandible length (posterior to anterior tip of the lower jaw) for all sampled specimens using the linear measurement tools in VG StudioMax and MeshLab (*Cignoni et al., 2008*). We calculated relative jaw length (mandible length divided by skull length) for each specimen: a jaw length value greater than 1 indicates a posteriorly shifted jaw joint (lower jaw is longer than the head) and a value less than 1 indicates an anteriorly shifted jaw joint (lower jaw is shorter than the head).

We used phylogenetic comparative methods to test for evolutionary correlations among dentition, diet, and body size in frogs. We compiled diet records for 268 taxa, representing 259 genera and 52 anuran families: 155 species in the dentition dataset had published diet records and the remaining 113 lineages are represented by genus-level diet data. We excluded the remaining 161 anuran species in the dentition dataset (55 edentulous, 106 toothed) from the diet analyses due to the lack of known diet records at the species or genus level. Because dentition (toothed/edentulous) and diet (generalist/microphagous) were treated as binary traits, we tested for a phylogenetic correlation using discrete independent and discrete dependent models with rjMCMC sampling in BayesTraits version 3.0.2 (*Pagel and Meade, 2006*). The stepping stone sampler for marginal likelihood reconstructions was used with 100 stones and 1000 iterations. The branch lengths were scaled to have a mean of 0.1 using ScaleTrees. Bayes factors (log BF = 2[log marginal likelihood complex model – log marginal likelihood simple model]) were used to compare the fit of the independent versus dependent models. Models were run using the complete 268 taxon dataset, a reduced 158 taxon dataset excluding genus-level diet data, and a reduced 134 taxon dataset excluding genus-level diet data and species-level diet data based on a small sample size (less than five individuals).

Several previous studies have demonstrated a correlation between skull shape and diet in frogs: species that specialize on small prey have anteriorly shifted, relatively short jaws while generalist feeders that are capable of eating large prey have a posteriorly shifted jaw joint (*Emerson, 1985*; *Vidal-García and Scott Keogh, 2017*; *Paluh et al., 2020*). Because diet data are lacking for many anuran genera, we additionally tested for a phylogenetic correlation between dentition and the relative length of the jaw as a morphological proxy for diet. Lastly, because teeth may be lost as a byproduct of miniaturization (*Hanken and Wake, 1993*; *Smirnov and Vasil'eva, 1995*), we tested for a phylogenetic correlation between dentition state and body size (SVL). Phylogenetic logistic regression models were calculated in the *phylolm* R package (*Ho and Ané, 2014*) using dentition and measurement data for 423 anuran species. Dentition (toothed/edentulous) was treated as the binary response variable and the log transformed size metrics (relative jaw length, SVL) as continuous predictor variables. We used the 'logistic_MPLE' method, which maximizes the penalized likelihood

of the logistic regression, with a btol of 10, a log.alpha.bound of 10, and 1000 bootstrap replicates. The body size analysis was run using the complete 423 taxon dataset and a reduced 377 taxon dataset excluding bufonid toads.

## Acknowledgements

We thank all of the institutions, curators, and collection managers that loaned us specimens for this study. We thank Marta Vidal-Garcia for providing access to the *Spicospina flammocaerulea* scan. We thank Kevin Conway for providing insight on the distribution of edentulism in actinopterygian fishes and David Wake for insights into the paravomerine teeth of salamanders. We thank the research group of DCB at the Florida Museum of Natural History and two anonymous reviewers for helpful comments that improved an earlier version of this manuscript.

## Additional information

### Funding

| Funder | Grant reference number | Author |
| --- | --- | --- |
| National Science Foundation | DGE-1315138 | Daniel J Paluh |
| National Science Foundation | DGE-1842473 | Daniel J Paluh |
| National Science Foundation | DBI- 1701714 | Edward L Stanley<br>David C Blackburn |

The funders had no role in study design, data collection and interpretation, or the decision to submit the work for publication.

### Author contributions

Daniel J Paluh, Conceptualization, Data curation, Formal analysis, Funding acquisition, Investigation, Visualization, Methodology, Writing - original draft, Writing - review and editing; Karina Riddell, Catherine M Early, Maggie M Hantak, Gregory FM Jongsma, Rachel M Keeffe, Fernanda Magalhães Silva, Stuart V Nielsen, María Camila Vallejo-Pareja, Data curation, Investigation, Writing - review and editing; Edward L Stanley, Resources, Data curation, Funding acquisition, Investigation, Writing - review and editing; David C Blackburn, Conceptualization, Resources, Data curation, Funding acquisition, Investigation, Writing - review and editing

### Author ORCIDs

Daniel J Paluh https://orcid.org/0000-0003-3506-2669
Maggie M Hantak https://orcid.org/0000-0001-9469-4741
Fernanda Magalhães Silva https://orcid.org/0000-0001-7741-7232
María Camila Vallejo-Pareja https://orcid.org/0000-0002-0987-3364
Edward L Stanley http://orcid.org/0000-0001-5257-037X
David C Blackburn https://orcid.org/0000-0002-1810-9886

### Decision letter and Author response

Decision letter https://doi.org/10.7554/eLife.66926.sa1
Author response https://doi.org/10.7554/eLife.66926.sa2

## Additional files

### Supplementary files

- Transparent reporting form

## Data availability

Computed tomography data (tiff stacks and mesh files) have been deposited in MorphoSource (see Dataset 1). Data and scripts for all analyses are available on GitHub at https://github.com/dpaluh/edentulous_frogs (copy archived at https://archive.softwareheritage.org/swh:1:rev:0a7b9b317ed7534536ab5e07d83a3fb4097842e0).

The following datasets were generated:

| Author(s) | Year | Dataset title | Dataset URL | Database and Identifier |
|---|---|---|---|---|
| Paluh DJ, Riddell K, Early CM, Hantak MM, Jongsma GF, Keeffe RM, Magalhães Silva F, Nielsen SV, Vallejo-Pareja MaC, Stanley EL, Blackburn DC | 2021 | Dataset S1 from: Rampant tooth loss across 200 million years of frog evolution | https://github.com/dpaluh/edentulous_frogs/blob/main/DatasetS1_dentition.xlsx | GitHub, dpaluh/edentulous_frogs/blob/main/DatasetS1_dentition.xlsx |
| Paluh DJ, Riddell K, Early CM, Hantak MM, Jongsma GF, Keeffe RM, Magalhães Silva F, Nielsen SV, Vallejo-Pareja MaC, Stanley EL, Blackburn DC | 2021 | Dataset S2 from: Rampant tooth loss across 200 million years of frog evolution | https://github.com/dpaluh/edentulous_frogs/blob/main/DatasetS2_diet.xlsx | GitHub, dpaluh/edentulous_frogs/blob/main/DatasetS2_diet.xlsx |
| Paluh DJ, Riddell K, Early CM, Hantak MM, Jongsma GF, Keeffe RM, Magalhães Silva F, Nielsen SV, Vallejo-Pareja MaC, Stanley EL, Blackburn DC | 2021 | Dataset S3 from: Rampant tooth loss across 200 million years of frog evolution | https://github.com/dpaluh/edentulous_frogs/blob/main/DatasetS3_measurements.xlsx | GitHub, dpaluh/edentulous_frogs/blob/main/DatasetS3_measurements.xlsx |

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
