## [Decision Letter]

**Acceptance summary:**

This manuscript will find a broad audience in the fields of herpetology, systematics, as well as those interested in the evolutionary history of vertebrate teeth. The expansive dataset presented by the authors has allowed for rigorous computational analyses yielding new insight into the evolutionary history of teeth in frogs, which is a topic that has received little attention from the scientific community. The resulting data largely support the key claims of the manuscript.

**Decision letter after peer review:**

Thank you for submitting your article "Rampant tooth loss across 200 million years of frog evolution" for consideration by *eLife*. Your article has been reviewed by 2 peer reviewers, and the evaluation has been overseen by Patricia Wittkopp as the Senior and Reviewing Editor. The reviewers have opted to remain anonymous.

Essential revisions:

Please address the suggestions from both reviewers provided below, which detail recommended changes to improve the presentation of this work.

*Reviewer #1 (Recommendations for the authors):*

Abstract: The term "broadly maintained" in the abstract is a bit vague. "Teeth are present in most…" is clearer.

Introduction:

Clear and concise. Although brief, I see no lack in critical information.

Results: There is a missed opportunity to examine the evolution of each regional tooth "type" in this study, which would greatly increase the study's impact and help draw conclusions about developmental hypotheses proposed in the discussion. For example, in the discussion it is mentioned that "Vomerine teeth are not coordinated with dentition on the upper jaw in frogs" and this could be tested and shown here. Are teeth types lost in any particular order? Or are other non-vomerine tooth types lost in a coordinated fashion? These analyses would be incredibly beneficial, even if just in supplemental materials.

Line 130: You define microphagy in the introduction but it would be ideal to specifically state in the Results section that microphagy was defined here as (> 50%) of their diet by number or volume consists of ants, termites, or mites.

Given the range of reliability in the diet data (e.g. "singular reports (one prey item in one individual) to detailed studies investigating the stomach contents of dozens of individuals through space and time"), it would be nice to see some attempted sensitivity analyses to better understand how species with single or very few prey records could be influencing your correlation.

Discussion:

Line 205: The relationship between the loss of mineralized structures and delayed ossification has been posited to explain the rampant loss of other skull structures frequently lost in anuran clades – most notably the loss of middle ear structures (see Pererya et al. 2016 Sci Reports, Womack et al. 2018 J Morph, Womack et al. 2019 Am Nat, and others). It would be worth mentioning this here, given it suggests this could be a more common phenomena.

Line 214 mentions that direct development may provide an opportunity to alter dental development but you provide no evidence that direct development is associated with shifts in dentition. This would be interesting and easy to examine with your current dataset. But if you feel it is beyond the scope of this paper, I would at least mention whether there appears to be a correlation based on existing data.

Line 287 the paragraph discussing regains does a great job pointing out that some teeth may be present but very small and that regains may or may not have occurred – we can never know based on these meshes alone. It would be worth adding that even if teeth are truly and completely lost in many microhylids, teeth could simply have been lost more times and never regained. This alternative scenario could be quantified by restricting ancestral state reconstructions to not allow regains, or simply stated as a possibility.

Line 319: Please change "despite that this new world radiation of over 1,000 species contains" to "despite this new world radiation of over 1,1000 species containing.."

Please also consider removing "new world" and "old world" from the manuscript. These terms, although widely used and accepted, are inherently colonial in their perspective.

The section on tooth loss in fossil frogs provides some really interesting and relevant information about tooth loss in this clade. It would be nice to see a bit of synthesis between the fossil and extant data (in the discussion), and/or some interpretation of the extant data collected and analyzed in the context of these fossil records. Are these crown-group fossil frogs just adding clades of tooth loss? Do they change any interpretations or assumptions about the analysis on extant taxa? As written, a general audience may struggle to make these connections if not familiar with these extinct and extant taxonomic groups.

Methods:

Is there any chance for error in dentition survey? Could teeth be present but very unmineralized and not recognized during segmentation?

You mention that some microhylids that would appear to lack teeth do have "true teeth" that are very small. How likely is this across frogs more generally? Is there histological evidence that shows a true lack of teeth in any species? This seems important to discuss more generally and not only in the context of microhylid regains.

Diet data are incredibly tough to attain from the literature for most amphibian species. The authors have done a great job and service compiling the data for this study. The authors point out the range in data that they drew their microphagy versus generalist categorizations from – "singular reports (one prey item in one individual) to detailed studies investigating the stomach contents of dozens of individuals through space and time". However, this range of data is really hard to gauge from the presented datasheet in the supplemental methods.

There are clear limitations from drawing diet categories from a singular report. I think it best to be transparent about your diet data and to more easily allow readers to evaluate and add to your dataset. My suggestion is to clearly state, not just the diet category and associated citation, but the corresponding amount and types of data found for each species – "how many individuals, how many/much prey, what was the number/proportion of prey types, etc". This will allow readers to: (1) assess which species have robust datasets that cover geographic and time sampling and which are based on singular or minimal reports and (2) more clearly identify the data that led the authors to the categorization of "microphagy" versus "generalist".

If this information is added to the publicly available dataset, it will greatly increase the impact of this paper on future amphibian diet research.

Was there any analysis of diet variation within genera to show that genus-level diet could be used as a proxy for species-level diet and endentulsim matched data? If endentulism (or lack thereof) is generally consistent within a genus and diet records are generally consistent among species in a genus, this seems like an ok assumption. I imagine genus level data could be data for one species in a genus or many. Responding to my comment above about more explicitly recording diet data would help readers assess.

*Reviewer #2 (Recommendations for the authors):*

1. Since this is one of the few models that I am aware of where teeth disappeared and reappeared on numerous occasions. In an extra "analysis", could the authors finding the estimated dates of the loss and subsequent reversal events? This would go a long way towards obtaining a better understanding of just how long teeth can be lost before they reappear. This would also provide some insight into the likelihood of these putative reversal events.

2. It may be helpful to highlight the major clades in Supplementary Figure 1 (in the periphery of the tree) in a similar manner to Figure 2.

3. Line 122 – 126. Would it be possible to make these groups (Mesobatrachia, Hyloidea, Ranoidea, etc…) in Figure 2 as has been done for the Families?

4. Line 122 – 126. Marking the reversal events with an asterisk in the tree would provide clarity.

5. Some of the scientific names in the legend for Figure 3 are not italicized

6. Line 125, the authors state "five reversals were inferred in Microhylidae (in Dyscophus, Uperodon, Anodonthyla, Cophyla, and Rhombophryne + Plethodontohyla). However, I cannot seem to find Dyscophus or Plethodontohyla within Supplementary Figure 1. Are they not in the tree?

Also, within the Microhylidae clade in Figure 2, since the above two species are missing, it appears that there are only 4 reversal events? in Anodonthyla, Cophyla, the two Rhombophryne species and the Uperodon.

7. In Figure 3, the labels for the skulls appear at first glance to be the binomial names, with the letter being the initial for the genus. However, this is clearly not the case. It may be a source of confusion for the reader. The authors may want to consider removing the letters if they don't serve an important purpose.

8. Lines 137 – 140 describe Bufonidae and Microhylidae as having edentulous and generalist feeders as members. However, there is some literature pointing to myrmecophagy in Bufonids.

Isacch, J. P., and Barg, M. (2002). Are bufonid toads specialized ant-feeders? A case test from the Argentinian flooding pampa. Journal of Natural History, 36(16), 2005-2012.

Teixeira, R. L., and Ferreira, R. B. (2009). Feeding pattern and use of reproductive habitat of the Striped toad Rhinella crucifer, Anura: Bufonidae, from Southeastern Brazil. Feeding pattern and use of reproductive habitat of the Striped toad Rhinella crucifer, Anura: Bufonidae, from Southeastern Brazil, 125-134.

9. First paragraph of Results section: I assume there were no palatine or pterygoid teeth in any frog? If not, it might be good to state this forthrightly (otherwise maybe state that "no other bones possessed teeth").

Discussion:

10. The Discussion is very broad overall, and while it is clearly well researched and provides lots of fascinating information, it is at times hard to follow and meanders in directions which may not necessarily be directly associated with the work here. For example, there is a section on dietary alkaloids that does not appear to have any bearing on tooth loss (at least that the authors connect together). I suggest that the authors focus the discussion more. While the information therein is very interesting for someone interested in amphibian biology, sometimes it's difficult to remember what the manuscript was about when reading it.

11. Line 60, the authors discuss the 22 independent tooth loss events, but could this have happened in association with anurans speciating much more than other dentulous groups? This idea may be worth discussing.

12. Could the larger bufonids have skewed the body size analysis? After all, tooth loss occurred in the common ancestor of bufonids, just once, and then some of these giant species may have arisen with no option but to be edentulous. In other words, would the data on body size and edentulism correlated significantly without the inclusion of bufonids?

---

## [Author Response]

Essential revisions:Please address the suggestions from both reviewers provided below, which detail recommended changes to improve the presentation of this work.Reviewer #1 (Recommendations for the authors):Abstract: The term "broadly maintained" in the abstract is a bit vague. "Teeth are present in most…" is clearer.

Change made.

Introduction:Clear and concise. Although brief, I see no lack in critical information.

Thank you.

Results: There is a missed opportunity to examine the evolution of each regional tooth "type" in this study, which would greatly increase the study's impact and help draw conclusions about developmental hypotheses proposed in the discussion. For example, in the discussion it is mentioned that "Vomerine teeth are not coordinated with dentition on the upper jaw in frogs" and this could be tested and shown here. Are teeth types lost in any particular order? Or are other non-vomerine tooth types lost in a coordinated fashion? These analyses would be incredibly beneficial, even if just in supplemental materials.

We agree that analysis of dental regions (beyond the Bayesian model-averaged ancestral states of tooth presence/absence on individual dentigerous elements that are provided in Figure 2—figure supplements 3–6) warranted further exploration. Thank you for the suggestion! Non-vomerine teeth in the frogs sampled on the premaxilla and maxilla are 100% coordinated in their presence/absence. We have rephased the following in the results to make this clear: “The evolution of maxillary and premaxillary teeth of the upper jaw is synchronized in all frog species (Figure 1A, 1B), being present in 292 taxa and coordinately absent in 136 species.” Therefore, we added an analysis testing if frog tooth loss is ordered, with vomerine tooth loss being an intermediate step between fully toothed (on the premaxilla, maxilla, vomer) and complete edentulism. This new analysis indicates that vomerine tooth loss does not always precede complete edentulism in frogs. We additionally inferred a regain of vomerine teeth in one species.

Results added:

“We compared six discrete character evolution models using fitMk in phytools (Revell 2012) for three dental states (fully toothed; toothed upper jaw with vomerine tooth loss; edentulous) in 425 anuran species to test if vomerine tooth loss precedes complete edentulism in frogs. […] Toothed frogs have lost vomerine teeth an estimated 59 times. One gain of vomerine teeth subsequent to a loss was inferred in *Phlyctimantis leonardi*.”

Discussion modified:

“The loss of vomerine teeth is not a prerequisite for complete tooth loss (Figure 3), and the presence or absence of teeth on the vomer is not coordinated with dentition on the maxilla and premaxilla in frogs. These results suggest that dentition on the upper jaws and palate are independent modules, and the higher lability of vomerine teeth requires further study.”

Methods added:

“To test if vomerine tooth loss always precedes complete edentulism in frogs, we compared discrete character evolution models using fitMk in phytools for three dental states (fully toothed; toothed upper jaw with vomerine tooth loss; edentulous) and 425 species of frogs. […] To estimate the posterior probabilities for all nodes and the number of transitions between these three character states, we used stochastic character mapping (MCMC sampling of character histories from their posterior probability distribution; Huelsenbeck et al. 2003) in phytools (make.simmap function) using the best fit model of character evolution (all-rates-different), a flat root prior, and 1000 replicates (Figure 3—figure supplement 2).”

Line 130: You define microphagy in the introduction but it would be ideal to specifically state in the Results section that microphagy was defined here as (> 50%) of their diet by number or volume consists of ants, termites, or mites.

Added “(defined here as > 50% of diet by number of volume consisting of ants, termites, or mites)”.

Given the range of reliability in the diet data (e.g. "singular reports (one prey item in one individual) to detailed studies investigating the stomach contents of dozens of individuals through space and time"), it would be nice to see some attempted sensitivity analyses to better understand how species with single or very few prey records could be influencing your correlation.

We added an additional "sensitivity" analysis removing taxa with diet data based on less than 5 specimens and the pattern of correlated evolution between edentulism and microphagy remains significant.

Modified methods:

“Models were run using the complete 268 taxon dataset, a reduced 158 taxon dataset excluding genus-level diet data, and a reduced 134 taxon dataset excluding genus-level diet data and species-level diet data based on a small sample size (less than 5 individuals).”

Modified Results:

“A BayesTrait discrete analysis indicated correlated evolution between edentulism and microphagy: the dependent model of trait evolution is strongly supported over the independent model (Bayes factor = 54.32; a Bayes factor > 2 implies the evolution of two traits is linked). Similar results were found using a 155-taxon dataset excluding genus-level diet data (Bayes factor = 26.28) and a 134-taxon dataset excluding genus-level diet data and species-level diet data based on a small sample size (less than 5 individuals; Bayes factor = 16.0).”

Discussion:Line 205: The relationship between the loss of mineralized structures and delayed ossification has been posited to explain the rampant loss of other skull structures frequently lost in anuran clades – most notably the loss of middle ear structures (see Pererya et al. 2016 Sci Reports, Womack et al. 2018 J Morph, Womack et al. 2019 Am Nat, and others). It would be worth mentioning this here, given it suggests this could be a more common phenomena.

Added:

“Truncated development is hypothesized to be associated with the repeated loss of other mineralized structures in frogs, such as the stapes of the middle ear (Pererya et al. 2016, Womack et al. 2018, Womack et al. 2019).”

Line 214 mentions that direct development may provide an opportunity to alter dental development but you provide no evidence that direct development is associated with shifts in dentition. This would be interesting and easy to examine with your current dataset. But if you feel it is beyond the scope of this paper, I would at least mention whether there appears to be a correlation based on existing data.

Thank you for pointing this out. We hypothesize direct development may alter aspects of dental development (such as timing of tooth germ initiation or tooth counts), but likely not complete edentulism based on our data. We have altered the following section to make this clear.

“Several anuran lineages have evolved direct development (undergoing the larval stage within the egg; Gomez-Mestre et al. 2012), and this life history transition may provide an opportunity to repattern the jaw and alter dental development, such as the timing of tooth germ initiation. Many of the edentulous frogs identified in this study are biphasic (possessing a free-swimming tadpole stage), but edentulism is also present in some direct-developing lineages (e.g., *Arthroleptis*, *Brachycephalus*, *Brevicipitidae*, *Myobatrachus*, asterophryine microhylids) and viviparous species (*Nimbaphrynoides*, *Nectophrynoides*), suggesting that life history mode does not constrain patterns of tooth loss.”

Line 287 the paragraph discussing regains does a great job pointing out that some teeth may be present but very small and that regains may or may not have occurred – we can never know based on these meshes alone. It would be worth adding that even if teeth are truly and completely lost in many microhylids, teeth could simply have been lost more times and never regained. This alternative scenario could be quantified by restricting ancestral state reconstructions to not allow regains, or simply stated as a possibility.

We agree and have added the following statement in this paragraph: “Alternatively, the most recent common ancestor of the Microhylidae possessed teeth and several lineages then subsequently lost dentition independently, with no reversals.”

Line 319: Please change "despite that this new world radiation of over 1,000 species contains" to "despite this new world radiation of over 1,1000 species containing.."

Change made.

Please also consider removing "new world" and "old world" from the manuscript. These terms, although widely used and accepted, are inherently colonial in their perspective.

Thank you for pointing this out, we have changed “new world” to “neotropical” (and recognize this is still not an ideal term)

The section on tooth loss in fossil frogs provides some really interesting and relevant information about tooth loss in this clade. It would be nice to see a bit of synthesis between the fossil and extant data (in the discussion), and/or some interpretation of the extant data collected and analyzed in the context of these fossil records. Are these crown-group fossil frogs just adding clades of tooth loss? Do they change any interpretations or assumptions about the analysis on extant taxa? As written, a general audience may struggle to make these connections if not familiar with these extinct and extant taxonomic groups.

We agree and have re-written the “fossil” Discussion section:

“Tooth loss in fossil amphibians

To our knowledge, no stem tetrapods have been described as edentulous (Ruta et al. 2003, Anderson et al. 2008, Matsumoto and Evans 2017). [...] The majority of living, edentulous frogs are neobatrachians (Figure 2; in Hyloidea, Ranoidea, Myobatrachidae, and Nasikabatrachidae), but few fossil neobatrachians have been described as edentulous (Gardner 2014).”

Methods:Is there any chance for error in dentition survey? Could teeth be present but very unmineralized and not recognized during segmentation?

Because true teeth are always densely mineralized (due to the presence of dentin and enamel), we are confident that our assessment of presence/absence using micro-CT data has a small chance of error. We have added the following statement in the methods to describe how we identified the presence of teeth:

“Teeth were identified by a combination of the following characteristics: conical shape, presence of distinct pedicel and crown, presence of replacement teeth, and/or presence of floating teeth that are undergoing resorption (Figure 1, Figure 2—figure supplement 1).”

You mention that some microhylids that would appear to lack teeth do have "true teeth" that are very small. How likely is this across frogs more generally? Is there histological evidence that shows a true lack of teeth in any species? This seems important to discuss more generally and not only in the context of microhylid regains.

We agree clarification was needed here, and so we modified the text and added a figure supplement. As mentioned above, because true teeth are always mineralized, it is evident if tooth-like structures are present or completely absent in CT data (as seen in *Gastrophryne* in Figure 1). Toothlessness in frogs has also been confirmed in several traditional morphological studies (a few examples include: *Rhinophrynus* [Trueb and Gans 1983]; *Gastrophryne* [Trueb et al. 2011]; some *Pipa* [Trueb and Cannatella 1986]).

The distinction between odontoids (bony serrations) and true teeth (with a pulp cavity, dentin, and enamel) can be more difficult to distinguish without histological data, and we now provide CT reconstructions of the microhylid species that have putatively regained teeth in Figure 2—figure supplement 1 to justify our hypotheses on true teeth vs. odontoids.

Modified text:

“If teeth were entirely lost in the common ancestor of microhylids, the repeated re-evolution of true teeth (with enamel, dentin, and pulp cavity) in this clade is unlikely and requires histological investigation. We hypothesize the tooth-like structures in some of these taxa, such as *Rhombophryne* and *Uperodon* (Figure 2—figure supplement 1), may be small odontoids (bony serrations that lack the tissue composition of true teeth; Fabrezi and Emerson 2003), […] The morphology of *Dyscophus*, *Anodonthyla*, *Cophyla*, and *Plethodontohyla* indicates that these taxa have true dentition based on the presence of replacement teeth and/or floating teeth that are undergoing resorption (Figure 2—figure supplement 1).”

Diet data are incredibly tough to attain from the literature for most amphibian species. The authors have done a great job and service compiling the data for this study. The authors point out the range in data that they drew their microphagy versus generalist categorizations from – "singular reports (one prey item in one individual) to detailed studies investigating the stomach contents of dozens of individuals through space and time". However, this range of data is really hard to gauge from the presented datasheet in the supplemental methods.There are clear limitations from drawing diet categories from a singular report. I think it best to be transparent about your diet data and to more easily allow readers to evaluate and add to your dataset. My suggestion is to clearly state, not just the diet category and associated citation, but the corresponding amount and types of data found for each species – "how many individuals, how many/much prey, what was the number/proportion of prey types, etc". This will allow readers to: (1) assess which species have robust datasets that cover geographic and time sampling and which are based on singular or minimal reports and (2) more clearly identify the data that led the authors to the categorization of "microphagy" versus "generalist".If this information is added to the publicly available dataset, it will greatly increase the impact of this paper on future amphibian diet research.

We agree and have added a “summary of diet” column in our diet dataset. Due to wide variation in the methods and reporting standards of diet studies (ranging from individual counts of prey items, % volume, % number, % mass, frequency of prey items across individuals, etc.), and lowest taxonomic rank reported (from phylum to species), the data are briefly summarized in free text format. This summary will allow readers to assess robustness of diet records and provide a justification for our categorization of “generalist” or “microphagy”. We added an additional "sensitivity" analysis removing taxa with diet data based on less than 5 specimens (see above).

Was there any analysis of diet variation within genera to show that genus-level diet could be used as a proxy for species-level diet and endentulsim matched data? If endentulism (or lack thereof) is generally consistent within a genus and diet records are generally consistent among species in a genus, this seems like an ok assumption. I imagine genus level data could be data for one species in a genus or many. Responding to my comment above about more explicitly recording diet data would help readers assess.

Based on the literature, tooth presence/absence and diet (at the level of generalist vs specialist) are generally consistent within a genus. Only ten of 461 frog genera are known to contain both edentulous and toothed species, but further intrageneric sampling is likely needed. No analysis was conducted to confirm genus-level diet data could be used as a proxy. We recognize this is a considerable assumption, which is why phylogenetic correlations were conducted including and excluding genus-level diet data. We have addressed the comment about diet records above.

Modified methods:

“For species with no published diet records, we searched for any existing diet records at the genus level because dentition state (toothed/edentulous) is generally consistent within a genus. Only ten of 461 anuran genera are known to contain both edentulous and toothed species (*Arthroleptis*, *Cacosternum*, *Engystomops*, *Glyphoglossus*, *Mini*, *Physalaemus*, *Pipa*, *Telmatobius*, *Uperodon*, *Uperoleia*).”

Reviewer #2 (Recommendations for the authors):1. Since this is one of the few models that I am aware of where teeth disappeared and reappeared on numerous occasions. In an extra "analysis", could the authors finding the estimated dates of the loss and subsequent reversal events? This would go a long way towards obtaining a better understanding of just how long teeth can be lost before they reappear. This would also provide some insight into the likelihood of these putative reversal events.

We agree that such an analysis would be interesting and provide insight on the timeframe and likelihood of putative losses and regains of teeth in microhylids. But as mentioned in the discussion, microhylid relationships (and divergence time estimates) have varied widely in recent studies, with divergence time estimates for the most recent common ancestor of Microhylidae (where we estimate teeth were lost) ranging from ~66 mya (Feng et al. 2017) to over 100 mya (Feng et al. 2017, Jetz and Pyron 2018, Streicher et al. 2020). Therefore, we have refrained from conducting such an analysis because the margin for error is high.

Modified: “The phylogenetic relationships and divergence time estimates among microhylid taxa remain controversial (Peloso et al. 2016, Feng et al. 2017, Streicher et al. 2020), which further impedes the interpretation of dental evolution in this group.”

2. It may be helpful to highlight the major clades in Supplementary Figure 1 (in the periphery of the tree) in a similar manner to Figure 2.

We agree; change made.

3. Line 122 – 126. Would it be possible to make these groups (Mesobatrachia, Hyloidea, Ranoidea, etc…) in Figure 2 as has been done for the Families?

Change made.

4. Line 122 – 126. Marking the reversal events with an asterisk in the tree would provide clarity.

Thank you for this suggestion, change made.

5. Some of the scientific names in the legend for Figure 3 are not italicized

Thank you, fixed.

6. Line 125, the authors state "five reversals were inferred in Microhylidae (in Dyscophus, Uperodon, Anodonthyla, Cophyla, and Rhombophryne + Plethodontohyla). However, I cannot seem to find Dyscophus or Plethodontohyla within Supplementary Figure 1. Are they not in the tree?

Thank you, fixed.

Also, within the Microhylidae clade in Figure 2, since the above two species are missing, it appears that there are only 4 reversal events? in Anodonthyla, Cophyla, the two Rhombophryne species and the Uperodon.

The asterisks now mark the 5 inferred reversals in Microhylidae in Figure 2—figure supplement 2.

7. In Figure 3, the labels for the skulls appear at first glance to be the binomial names, with the letter being the initial for the genus. However, this is clearly not the case. It may be a source of confusion for the reader. The authors may want to consider removing the letters if they don't serve an important purpose.

We agree this could be potentially confusing and so have replaced the letters with numbers. These serve the important purpose of indicating which tree tips correspond to each skull.

8. Lines 137 – 140 describe Bufonidae and Microhylidae as having edentulous and generalist feeders as members. However, there is some literature pointing to myrmecophagy in Bufonids.Isacch, J. P., and Barg, M. (2002). Are bufonid toads specialized ant-feeders? A case test from the Argentinian flooding pampa. Journal of Natural History, 36(16), 2005-2012.Teixeira, R. L., and Ferreira, R. B. (2009). Feeding pattern and use of reproductive habitat of the Striped toad Rhinella crucifer, Anura: Bufonidae, from Southeastern Brazil. Feeding pattern and use of reproductive habitat of the Striped toad Rhinella crucifer, Anura: Bufonidae, from Southeastern Brazil, 125-134.

We have modified Lines 137 – 140 for clarity:

“The majority of the 26 taxa classified as both edentulous and generalist feeders are a subset of the bufonids and microhylids sampled, but also includes the fully aquatic pipids…”

This is also discussed in the discussion (line 284):

“Once lost, teeth have not been regained in the Bufonidae but may have re-evolved several times in microhylids. […] The variation in diet within bufonids and microhylids corresponds with variation in the relative length of the lower jaw (Dataset S3) and overall skull morphology (Paluh et al. 2020).”

9. First paragraph of Results section: I assume there were no palatine or pterygoid teeth in any frog? If not, it might be good to state this forthrightly (otherwise maybe state that "no other bones possessed teeth").

This is a good point, thank you. We added “No other anuran cranial elements possess teeth.”

Discussion:10. The Discussion is very broad overall, and while it is clearly well researched and provides lots of fascinating information, it is at times hard to follow and meanders in directions which may not necessarily be directly associated with the work here. For example, there is a section on dietary alkaloids that does not appear to have any bearing on tooth loss (at least that the authors connect together). I suggest that the authors focus the discussion more. While the information therein is very interesting for someone interested in amphibian biology, sometimes it's difficult to remember what the manuscript was about when reading it.

While we appreciate this concern, comments from Reviewer 1 pointed us in a different direction that led to expanding our discussion further. Given that nearly none of this has ever been summarized in the literature, and that there are no page limits on manuscripts in *eLife*, we feel our discussion is clearly of value for our community interested in amphibian biology, as indicated by comments from Reviewer 1. For example, nearly all origins of dietary alkaloid sequestration in frogs co-occur with microphagy and edentulism, although a few exceptions exist as noted in the discussion. The ecology and evolution of alkaloid sequestration and diet in toxic frogs is studied extensively by several research groups, and the repeated loss of teeth in these clades is yet another convergent trait observed across these lineages (in addition to aposematic color, diurnal activity, locomotion, breeding biology, etc.).

11. Line 60, the authors discuss the 22 independent tooth loss events, but could this have happened in association with anurans speciating much more than other dentulous groups? This idea may be worth discussing.

This is an interesting idea that could be tested using trait-dependent diversification models across all major vertebrates, but we feel it is beyond the scope of this paper. We predict that speciation rates and edentulism are not correlated, however. For example, there are only three known cases of edentulism in Actinopterygii and zero cases in Squamata, two clades with higher species diversity than frogs.

12. Could the larger bufonids have skewed the body size analysis? After all, tooth loss occurred in the common ancestor of bufonids, just once, and then some of these giant species may have arisen with no option but to be edentulous. In other words, would the data on body size and edentulism correlated significantly without the inclusion of bufonids?

This is a good point. We added an additional phylogenetic logistic regression analysis testing for correlated evolution between edentulism and SVL excluding bufonid toads and found a significant relationship. The following text was added:

Results:

“However, a relationship was found between edentulism and body size when excluding bufonid toads (alpha = 0.0020, standard error = 0.1539, *P* = 0.015), a clade representing a single loss of teeth but that vary widely in size (11.7–152.4 mm SVL in our dataset).”

Discussion:

“When excluding bufonids, reduction in body size was phylogenetically correlated with edentulism, indicating that further work is needed to investigate the interplay between edentulism, body size evolution, and paedomorphism across anurans.”

Methods:

“The body size analysis was run using the complete 423 taxon dataset and a reduced 377 taxon dataset excluding bufonid toads.”